# ZDHHC19 Is Dispensable for Spermatogenesis, but Is Essential for Sperm Functions in Mice

**DOI:** 10.3390/ijms22168894

**Published:** 2021-08-18

**Authors:** Shuai Wang, Hongjie Qiao, Pengxiang Wang, Yuan Wang, Danian Qin

**Affiliations:** 1Department of Physiology, Shantou University Medical College, Shantou 515041, China; wangshuai2509@126.com (S.W.); 13hjqiao@stu.edu.cn (H.Q.); 2Shanghai Key Laboratory of Regulatory Biology, Institute of Biomedical Sciences and School of Life Sciences, East China Normal University, Shanghai 200241, China; wangpengxiang1900@163.com; 3Department of Animal Sciences, College of Agriculture and Natural Resources, Michigan State University, East Lansing, MI 48824, USA

**Keywords:** *Zdhhc19*, spermatogenesis, fertility, sperm motility, acrosome reaction

## Abstract

Spermatogenesis is a complicated process involving mitotically proliferating spermatogonial cells, meiotically dividing spermatocytes, and spermatid going through maturation into spermatozoa. The post-translational modifications of proteins play important roles in this biological process. *S*-palmitoylation is one type of protein modifications catalyzed by zinc finger Asp-His-His-Cys (ZDHHC)-family palmitoyl *S*-acyltransferases. There are 23 mammalian ZDHHCs that have been identified in mouse. Among them, *Zdhhc19* is highly expressed in adult testis. However, the in vivo function of *Zdhhc19* in mouse spermatogenesis and fertility remains unknown. In this study, we knocked out the *Zdhhc19* gene by generating a 2609 bp deletion from exon 3 to exon 6 in mice. No differences were found in testis morphology and testis/body weight ratios upon *Zdhhc19* deletion. Spermatogenesis was not disrupted in *Zdhhc19* knockout mice, in which properly developed TRA98+ germ cells, SYCP3+ spermatocytes, and TNP1+ spermatids/spermatozoa were detected in seminiferous tubules. Nevertheless, *Zdhhc19* knockout mice were male infertile. *Zdhhc19* deficient spermatozoa exhibited multiple defects including abnormal morphology of sperm tails and heads, decreased motility, and disturbed acrosome reaction. All of these led to the inability of *Zdhhc19* mutant sperm to fertilize oocytes in IVF assays. Taken together, our results support the fact that *Zdhhc19* is a testis enriched gene dispensable for spermatogenesis, but is essential for sperm functions in mice.

## 1. Introduction

Spermatogenesis is a complex developmental process, during which spermatogonial stem cells (SSCs) undergo self-renewal and differentiation into spermatocytes and then generate haploid spermatids through two successive meiosis. This highly orchestrated process is controlled by intricate molecular programs including translational and post-translational regulations. Disturbing spermatogenesis will lead to male infertility [1].

Multiple protein post-translational modifications (PTMs) occur in germ cells during spermatogenesis, such as acetylation [2,3], crotonylation [4], glycosylation [5,6,7], phosphorylation [8,9,10], SUMOylation [11], and ubiquitination [12]. Many of these PTMs have been identified to play important roles in sustaining mammalian spermatogenesis and male fertility [13,14]. For example, acetylated α-TUBULIN is reduced in individuals with poor sperm motility [15]. Chromodomain protein CDYL regulates histone Crotonylation in spermatogenesis. *Cdyl* transgenic mice manifest dysregulation of histone lysine crotonylation and reduction of male fertility with a decreased epididymal sperm count and sperm cell motility [16]. α-mannosidase IIx (MX) is an enzyme involved in *N*-glycan synthesis of proteins. Upon *MX* gene knockout, mice demonstrate male infertility due to reduced GlcNAc-terminated complex type *N*-glycans [17]. Src homology domain tyrosine phosphatase 2 (SHP2) is a tyrosine phosphatase, depletion of *Shp2* in germ cell causes meiotic spermatocytes to die and leads to sterility in mice [18]. Ubiquitination-deficient mutations in P-element Induced Wimpy Testis (PIWI)*,* a key protein for germ cell development, cause male infertility by impairing histone-to-protamine exchange during spermiogenesis [19]. All these studies thus support that proper PTMs are required for maintaining spermatogenesis and male fertility in mice.

Palmitoylation is one type of PTMs, in which proteins are modified with palmitic acids (sixteen-carbon saturated fatty acid) [20,21]. Palmitoylation can occur through *N*-palmitoylation, *O*-palmitoylation, and *S*-palmitoylation. *S*-palmitoylation is the thioesterification that occurs on an internal cysteine residue and is more often found in palmitoylated proteins [22]. *S*-palmitoylation is a reversible post-translational modification catalyzed by zinc finger Asp-His-His-Cys (ZDHHC)-family palmitoyl *S*-acyltransferases. By contrast, protein *N*-palmitoylation is essentially irreversible [22]. To date, 23 mammalian ZDHHCs have been identified [23,24]. Most ZDHHC proteins are localized to the Golgi and Endoplasmic Reticulum (ER) apparatus [25,26]; however, ZDHHC5, ZDHHC20, and ZDHHC21 are commonly localized to the cell membrane [27]. *S*-Palmitoylation plays a crucial role in cell signaling, protein localization, and protein-protein interactions [20,21,28,29].

Ohno et al. investigated the ZDHHCs genes expression patterns in human by RT-PCR, and classified ZDHHC genes into roughly three groups according to their tissue-specific expression patterns: highly ubiquitous, nearly ubiquitous, and tissue-specific [27]. The highly ubiquitous and nearly ubiquitous groups each include most of ZDHHC genes (ZDHHC4, 5, 7, 8, 10, 12, 13, 17, and 22; and ZDHHC1, 3, 6, 9, 14, 16, 18, and 21, respectively), indicating that multiple ZDHHC proteins are co-expressed in the same cell. Interestingly, a small number of ZDHHC genes are tissue-specific, including ZDHHC2, 11, 15, 19, and 20. *Zdhhc11* and *Zdhhc19* mRNAs were unique to testis [27]. We therefore decided to investigate the functions of the *Zdhhc19* gene in spermatogenesis by generating *Zdhhc19* knockout mice with the CRISPR/Cas9 technology. Male mice lacking *Zdhhc19* expression had normal spermatogenesis. However, their fertility was drastically affected. Anomalies were observed in *Zdhhc19* deficient sperm, both at the structural and functional levels.

## 2. Results

### 2.1. Zdhhc19 mRNA Is Highly Expressed in Testis

To investigate the role of ZDHHC19 in mouse testis, we first examined *Zdhhc19* transcript levels in multiple tissues from adult mice by qPCR. Our results showed that *Zdhhc19* mRNA was abundantly expressed in testes, but not detected in other tissues (Figure 1A,C). Because *Kit^w/wv^* mice lack of endogenous spermatogenesis, only supporting cells (e.g., Sertoli cells, Leydig cells, vascular endothelial cells, and fibroblast, etc.) are present in adult mouse testes. We thus collected testes from *Kit^w/wv^* mice to exclude the possibility that ZDHHC19 is expressed in testicle somatic cells. Indeed, no visible *Zdhhc19* transcripts were detected from *Kit^w/wv^* testes (Figure 1A,C). In addition, we found gradually increased *Zdhhc19* mRNA expression in testes at postnatal 1-, 2-, 3-, 4-, and 5-week, with the highest level at 5-week (Figure 1B). As meiosis starts in mice around postnatal day 8, these data suggest a potential role of ZDHHC19 in germ cell development at the post-meiosis stage.

### 2.2. Generation of Zdhhc19 Knockout (KO) Mice

To analyze the physiological functions of *Zdhhc19* gene, we utilized the CRISPR/Cas9 technique to establish the *Zdhhc19* knockout (KO) mouse model. Homozygous mutant mice were generated and validated by PCR and Sanger sequencing, containing a 2609 bp deletion from exon 3 to 6, which led to frame-shift mutations of *Zdhhc19* gene (Figure 2A,B). In addition, qPCR analysis demonstrated that *Zdhhc19* mRNA was hardly detected in the testes of *Zdhhc19* KO mice (Figure 2C), suggesting that *Zdhhc19* gene was successfully knocked out. No obvious developmental abnormalities were observed in the *Zdhhc19*^−/−^ mouse lines compared with *Zdhhc19*^+/+^ and *Zdhhc19*^+/−^ littermates.

### 2.3. Zdhhc19 Deletion Does Not Affect Spermatogenesis

The deletion of the *Zdhhc19* gene could offer an opportunity to understand the functional contribution of *Zdhhc19* to mouse spermatogenesis and male fertility. We thus investigated the requirement for *Zdhhc19* in mouse spermatogenesis using this *Zdhhc19* knockout mouse model.

We dissected the testes of control and KO mice at 56 dpp, and found the testis in KO mice was morphologically normal (Figure 3A); the testicular size and testes/body weight ratio of KO mice were comparable to their littermate controls at 42 dpp (Figure 3A,B). Postnatal germ cell development in control and KO mice were then examined with immunohistofluorescence (IHF). SYCP3 is a spermatocyte-specific marker [30], and its signals were comparable in KO and control mice at 42 dpp (Figure 3C), indicating a normal spermatocyte development. TRA98 is a pan-germ cell marker [31], and its staining revealed normal meiosis and haploid germ cell development in KO mice at 42 dpp (Figure 3C). TNP1, a protein that only marks haploid germ cells [32], was also readily detected in testis sections from *Zdhhc19* KO mice, indicating the existence of spermatids and spermatozoa in the inner adluminal compartments of seminiferous tubules, similar to those in control littermates (Figure 3D).We further analyzed the epididymides and found no difference in the numbers of spermatozoa in *Zdhhc19* KO mice compared to controls (Figure 3E). Finally, we evaluated the impact of *Zdhhc19* deletion on male fertility using a breeding test. We found that the breeding performance of *Zdhhc19* KO males were significantly reduced. Breeding of three *Zdhhc19* KO males with wild-type females over a three-month period yielded no offspring (Figure 3F). By contrast, wildtype controls gave an average approximately six pups per litter (Figure 3F). In summary, we conclude that *Zdhhc19* is dispensable for spermatogenesis, but is essential for sustaining male fertility in mice.

### 2.4. Knockout of Zdhhc19 Gene Leads to Male Infertility by Affecting Sperm Motility

Spermatozoa must undergo functional maturation in the epididymis after leaving testis before they can competently interact with oocytes [33]. This process includes the acquisition of sperm motility and the potential to go through the sperm capacitation and acrosome reaction. To understand the underlying causes of male infertility of *Zdhhc19* KO mice, we assessed the motility of sperm isolated from the epididymis by CASA. As shown in Figure 4, the sperm motility that was measured by main motility parameters of total motility (Figure 4A) and curvilinear velocity (VCL) (Figure 4B) were significantly decreased upon *Zdhhc19* deletion. However, other velocity parameters including straight line velocity (VSL) and average path velocity (VAP) were comparable in KO and control sperm (Figure 4B). In addition, amplitude of lateral head displacement (ALH) and beat/cross frequency (BCF) were both significantly reduced in *Zdhhc19* KO mice (Figure 4C,D).

To further understand whether the structure of sperm in *Zdhhc19* KO mice was altered, we collected spermatozoa from cauda epididymides and conducted immunofluorescence (IF) analyses with an antibody against acetylated-TUBULIN, which recognize the antigen in sperm flagella. We found various morphological abnormalities in *Zdhhc19* KO sperm tails, including coiled (arrowhead) and bent (arrow) sperm flagella, whereas sperm from control littermates developed normally (Figure 4E).

We then examined the structure of the sperm heads using IF assays with an acetylated-TUBULIN antibody and PNA staining. We did not find obvious abnormalities in heads of *Zdhhc19* KO sperm (Appendix A). However, CASA quantification revealed a higher width/length ratio (Elongation) and a smaller square of sperm head (Area) in *Zdhhc19* KO sperm, compared with control sperm (Figure 4F,G), indicating that the loss of *Zdhhc19* also causes abnormalities in sperm head development. The exact abnormalities of sperm heads upon *Zdhhc19* KO may require ultrastructural analyses using transmission electron microscopy. Taken together, despite the fact that *Zdhhc19* is not required for spermatogenesis, *Zdhhc19* deficiency causes structural abnormalities in sperm tails and probably in sperm heads as well.

### 2.5. Zdhhc19 Is Dispensable for Acrosome Biogenesis but Is Required for the Acrosome Reaction

The acrosome is a specialized organelle that covers the anterior part of the sperm head and contains hydrolytic enzymes that are essential for acrosome reaction to help sperm penetrate the zona pellucida (ZP) of oocytes. Therefore, properly formed acrosome and appropriate acrosome reaction play critical roles in mammalian fertilization [34]. Because *Zdhhc19* KO mice are infertile with possible abnormalities in sperm heads, we sought to investigate whether *Zdhhc19* is required for acrosome biogenesis and proper acrosome reaction.

We performed immunostaining assays with Rhodamine-PNA (red). In this assay, intact acrosomes are labeled with Rhodamine-PNA, whereas sperm that had undergone acrosome reaction will be stained negative. As shown in Figure 5A, the acrosomes in sperm from both KO and control males were formed properly, as labeled with the red fluorescent Rhodamine-PNA dye (Figure 5A). About 18% of sperm from *Zdhhc19* KO mice underwent spontaneous acrosome reaction, similar to that of control littermates (Figure 5B). When treated with A23187, a commonly used chemical that induces the acrosome reaction in vitro, more than 40% of sperm from control mice responded well and underwent acrosome reaction (Figure 5A,B). By contrast, the ratio of Rhodamine-PNA negative sperm remained around 18% upon *Zdhhc19* deletion (Figure 5A,B). These results suggest that *Zdhhc19* is essential for induced acrosome reaction, but dispensable for acrosome biogenesis.

### 2.6. Zdhhc19 Is Essential for In Vitro Fertilization in Mice

Given that *Zdhhc19* KO mice were sterile yet with normal spermatogenesis, reduced sperm motility, and abnormal acrosome reaction, male infertility can be caused by failure of spermatozoa to migrate through the female uterotubal junction (UTJ) due to low sperm motility [35]. Alternatively, the sterility of *Zdhhc19* KO mice may be attributed to defects in induced acrosome reaction and oocyte interaction/activation. To further distinguish the potential causes for male infertility upon *Zdhhc19* KO, we performed the in vitro fertilization (IVF) assay, which can exclude the contribution of failure in UTJ migration of *Zdhhc19* KO sperm. We used cumulus-intact eggs and monitored their fertilization by sperm to become two-cell embryos. Interestingly, we found a dramatically lower percentage (6/127 KO vs. 74/136 WT) of two-cell stage embryos after IVF with *Zdhhc19* KO sperm, compared to that of wild type sperm (Figure 6A,B).

## 3. Discussion

Gene-expression analysis studies estimate that more than 2300 genes in the mouse genome are expressed predominantly in the male germ line [36]. These testis-enriched genes are thought to play important roles in spermatogenesis and fertilization [37]. Indeed, some testis specific genes were found to be necessary for spermatogenesis and male fertility, such as the testis expressed gene 14 (TEX14) [38], the oocyte fusion factor IZUMO1 [39], and the component of the sperm flagellar dynein regulatory complex TCTE1 [40]. However, Ikawa et al. reported about 100 evolutionarily conserved and testis enriched genes were dispensable for male mouse fertility [37,41,42]. Up to now, many testes specific genes remain unexplored for their exact roles in spermatogenesis and male reproduction. Gene knockout is the “gold standard” to determine the physiological roles of a gene in vivo. With the development of the CRISPR/Cas9 gene editing technique, it is now possible to rapidly generate a knockout mouse model in a cost-effective manner. In this study, we established *Zdhhc19* KO mice by the CRISPR/Cas9 system to study its roles in spermatogenesis.

*Zdhhc19* is a testis-enriched gene that is highly conserved in all the eutherians from yeast to mammal [27]. The protein encoded by *Zdhhc19* consists of zinc finger DHHC–Cys-rich domains which are the catalytic region for its palmitoyl *S*-acyltransferase activity [22,27]. The expression profile of *Zdhhc19* in mouse tissues shows that *Zdhhc19* transcript is predominantly present in mouse testes (Mouse ENCODE transcriptome data: https://www.ncbi.nlm.nih.gov/gene/245308, updated on 23 June 2021). We also found *Zdhhc19* mRNAs mainly in postnatal germ cells, suggesting that *Zdhhc19* plays potential roles in regulating spermatogenesis and/or fertility in mice. However, the specific types of germ cells (i.e., spermatogonia, spermatocyte, and haploid spermatid) in which ZDHHC19 is expressed in the testis are not known due to lack of commercial antibodies against this protein. To investigate the exact expression pattern of ZDHHC19 in the testis, specific antibodies for IHF should be developed. The transgenic reporter mice by knocking in a GFP cDNA under a *Zdhhc19* promoter may also be used.

After successfully generating the *Zdhhc19* KO mouse model, we examined postnatal germ cell development upon loss of function of *Zdhhc19*. We found that spermatocytes and spermatids developed normally in *Zdhhc19* KO mice, which are however male infertile. One possibility of such phenomena may be due to the functional redundancy of other ZDHHC family members. For example, *Zdhhc11*, which is also expressed in mouse testis (our unpublished data), may work to compensate for the loss of function of *Zdhhc19* in early spermatogenesis. Alternatively, we found that *Zdhhc19* expression increased during spermatogenesis and peaked at postnatal week 4–5 (Figure 1B), when elongating or elongated spermatids first appear in the testis [43]. Therefore, ZDHHC19 may have a pivotal role in forming functional spermatozoa/sperm in the epididymis, rather than spermatogenesis in the testis.

Sperm motility is highly related to their tail structures and functions of sperm tail proteins [44]. Indeed, our sperm morphology analysis showed that *Zdhhc19* KO sperm had coiled and bent tails (Figure 4E). Under natural fertilization conditions, sperm must migrate through the UTJ, penetrate into the zona pellucida, and fuse to the oocyte [35]. Although spermatozoa with low motility may not reach UTJ, it is possible that they are capable of penetrating eggs in vitro. We thus performed IVF assay, and found that *Zdhhc19* KO spermatozoa with low motility could not fertilize intact oocytes, indicating that the reduced sperm motility only partially accounts for the sterility in KO mice. Because *Zdhhc19* KO spermatozoa contained potential defects in their head formation (Figure 4F,G) and did not respond well to A23187 (Figure 5A,B), it is plausible that impaired ability of *Zdhhc19* KO sperm in induced acrosome reaction is a primary cause for their male infertility.

The exact molecular mechanism by which ZDHHC19 regulates sperm functions remains unknown. Palmitoylation has been shown to be required for protein stability, localization, enzymatic activities, and protein–protein and protein–lipid interactions [27,29,45]. As a palmitoyl *S*-acyltransferase, ZDHHC19 likely acts through its substrates to regulate sperm functions. Currently, the protein targets of ZDHHC19 in germ cells are yet to be discovered. Given that the peak expression of ZDHHC19 occurs in the late stage of spermatogenesis, we speculate that ZDHHC19 acts upon de novo synthesized motor proteins and/or structure-related cytoskeleton proteins for palmitoylation during epididymal maturation, which in turn impact acrosome reaction, sperm tail/head development, and sperm mobility. Future studies are warranted to identify the specific palmitoylated substrates by ZDHHC19 to further elucidate its molecular mechanism in sustaining sperm functions.

## 4. Materials and Methods

### 4.1. Mouse Lines, Animal Care, and Fertility Test

*Zdhhc19* knockout mice were generated in Cyagen Biosciences (Suzhou, China) using the CRISPR/Cas9 techniques. The strategy for gene targeting is shown in Figure 2A. The target DNA sequences with PAM were: gRNA-1-AATGCGCAGCCATAGAATCCTGG and gRNA-2-GTGTGGTCCACGGTGTTAGCAGG. The sequence primer was: GTGTTTGCTGCCTTCAATGTAACG. For the generation of *Zdhhc19* KO mice, the female and male founders carrying a heterozygous (+/−) deletion mutation were inbred to produce homozygous (−/−) mice. The progeny mice were genotyped with PCR. For the fertility test, three adult KO and control males were separately paired with two sexually mature wild-type females for at least 2 months. The number of pups were counted and recorded. Pups per litter were presented as the number of total pups born divided by the number of litters. All animal experimental procedures (Protocol ID: AR201305007) were conducted in accordance with the local Animal Welfare Act and Public Health Service Policy (consistent with the WMA Statement on Animal Use in Biomedical Research) and approved on 1 May 2013 by the Committee of Animal Experimental Ethics at East China Normal University.

### 4.2. Genomic DNA Extraction and Genotyping

Genomic DNA was isolated as described previously [46]. Briefly, tail tips were digested in buffer A at 95 °C for 30 min, and the reactions were then stopped by adding buffer B. The supernatant was collected for PCR genotyping with Taq Master Mix (Vazyme Biotech, Nanjing, China, P112-AA) according to the manufacturer′s instructions. Sequences of primers and size of products are provided in Appendix A.

### 4.3. Total RNA Extraction, Reverse Transcription, and Real-Time PCR

As previously described [46], total RNA was extracted with RNAiso Plus solution (TAKARA, Dalian, China, 9109), and cDNAs were synthesized using a PrimeScrip RT reagent Kit (TAKARA, RR037A). Quantitative Real-time PCR was performed on Thermo Scientific QuantStudio 3 Real-Time PCR System using FastStart Universal SYBR Green Master (Roche Life Science, Mannheim, Germany, 04913914001). The data were analyzed using the comparative threshold cycle (ΔΔCt) method and normalized to *Gapdh*. All PCR primers used are listed in Appendix A.

### 4.4. Histology and Immunohistofluorensce (IHF)

IHF was performed as described previously [46]. Testes were fixed in 4% paraformaldehyde (PFA) at 4 °C overnight and embedded in paraffin. Antibodies used in this study, TRA98 (ab82527; 1:500), SYCP3 (ab15093; 1:400), and TNP1 (ab73135; 1:1000), were purchased from Abcam (Cambridge, MA, USA). Alexa Fluor^®^ conjugated secondary antibodies came from Jackson ImmunoResearch Laboratories (West Grove, PA, USA). Nuclei were counterstained with DAPI/ProLong™ Diamond Antifade Mountant (Thermo Fisher Scientific, Eugene, OR, USA, P36966). All images were collected using an Olympus BX53 microscope system (Olympus Life Science, Japan), and Brightness/Contrast and Channel Merges of pictures were processed with Image J software.

### 4.5. Immunofluorescence Staining (IF)

Sperm was collected after swimming out from the cauda epididymides into1x PBS, then were mounted on glass slides and dried. Slides were fixed in 4% paraformaldehyde (PFA) and permeabilized with 0.2% Triton X-100. After washing with PBS, slides were blocked by 10% Fetal Bovine Serum (FBS) (in 1× PBS with 0.5% Tween-20) at room temperature for 1 h, followed with primary antibodies incubation at 4 °C overnight, PBST (1× PBS with 0.1% Tween 20) washed sperm sections for 5 min thrice, incubated with secondary antibodies (in 1× PBS with 1% BSA) for 1 h at room temperature, counterstained with DAPI/ProLong™ Diamond Antifade Mountant. Immunofluorescence Images were captured and observed by Olympus BX53 microscope system, and processed with Image J. An antibody in this study, Acetylated TUBULIN (ab179484; 1:500) was purchased from Abcam. Alexa Fluor^®^ conjugated secondary antibodies came from Jackson ImmunoResearch Laboratories.

### 4.6. Sperm Motility Analysis

Sperm motility analysis was performed following a published protocol with minor modification [47]. Briefly, three KO and control male mice (12–16 weeks old) were sacrificed by cervical dislocation and sperm from the epididymis were collected and dispersed in 1 mL of 1× PBS. After incubation for 20 min at 37 °C in air, a 20-μL aliquot of the sperm suspension was then placed in a counting chamber (Leja SC100-01-02-A-CE) for assessment of motility by using an HTM-TOX IVOS sperm motility analyzer (version 14, Hamilton Thorne Biosciences). The parameters assayed were the percentage of motile sperm cells, Average Path Velocity (VAP), Straight-line Velocity (VSL), Curvilinear Velocity (VCL), Amplitude of Lateral Head displacement (ALH), and Beat Cross Frequency (BCF).

### 4.7. In Vitro Fertilization in Mice (IVF)

In vitro fertilization was performed as previously described with minor modifications [34]. Mature C57/B6L female mice were injected with pregnant mare serum gonadotropin (PMSG) (Ningbo Sansheng Pharmaceutical Co., Ltd., Ningbo, China) followed by Human Chorionic Gonadotropin (hCG) (Ningbo Sansheng Pharmaceutical co., Ltd., China) after 48 h. The female mice were then euthanized by cervical dislocation, after injecting hCG 13 h. Oviducts were collected in a 35-mm dish containing HTF medium (Nanjing Aibei Biotechnology Co., Ltd., Nanjing, China, M1130), and stage MII oocytes were collected from the oviducts. Mature sperm were collected from the cauda epididymis and transferred into the HTF medium for capacitation at 37 °C in a humidified incubator with 5% CO_2_, 95% air. After 30–60 min, the capacitated sperm were added to the fertilization droplet containing the oocytes. After 4–6 h of incubation, the oocytes were transferred to KSOM medium. The IVF rate was measured based on the proportion of 2-cell embryos at 24 h after insemination.

### 4.8. Acrosome Reaction Analysis

Mature sperm from the cauda epididymis were incubated in HTF medium at 37 °C in a humidified incubator with 5% CO2 and 95% air to allow capacitation. After 1 h, the calcium ionophore A23187 (MedChemExpressm LLC, Shanghai, China, HY-N6687) was added at final concentration of 20 μm to induce the acrosome reaction. After an additional 1h incubation, sperm were smeared on a glass microscope slide, dried at room temperature, and fixed with methanol at 20 °C for 30 s. Intact acrosomes were stained with Rhodamine-PNA (Vector Laboratories, RL-1072; 1:1000), and the sperm nuclei were labeled with DAPI. The acrosome status was evaluated by staining with Rhodamine-PNA, which binds the outer acrosomal membrane and therefore would not stain acrosome-damaged and acrosome-reacted sperm. More than 200 sperm were examined for all experimental conditions [48].

### 4.9. Statistics

The data were evaluated for significant differences using Student’s *t*-test and the 2-way ANOVA, calculated with GraphPadPrism5 software (GraphPad Software, La Jolla, CA, USA). A *p*-value < 0.05 was considered statistically significant.

## Figures and Tables

**Figure 1 ijms-22-08894-f001:**
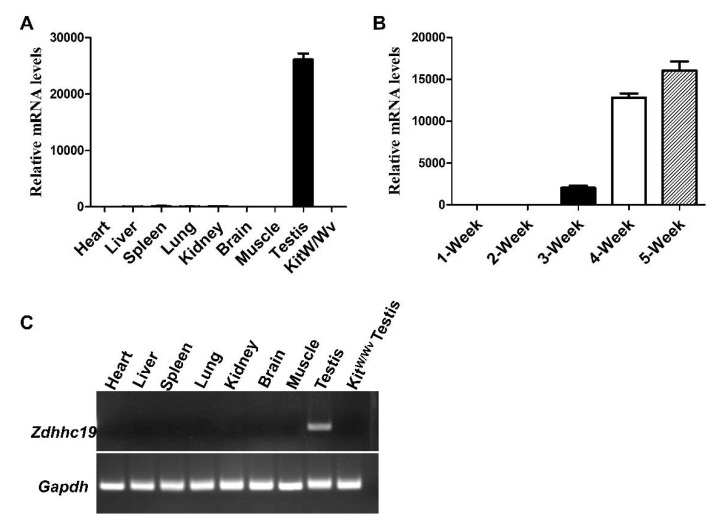
*Zdhhc19* is highly expressed in testicular germ cells but not in other tissues. (**A**) qPCR analyses of *Zdhhc19* mRNA expression levels in eight different organs from wildtype adult mice and Sertoli cells (*Kit^W/Wv^* testis). (**B**) qPCR analyses of *Zdhhc19* mRNA expression levels in developing testes at 1-, 2-, 3-, 4-, and 5-week. (**A**,**B**) *Zdhhc19* mRNA expression level in heart (**A**) and 1-Week (**B**) were used as baseline control, *Gapdh* served as the cDNA loading control. Data are presented as mean ± SEM. (**C**) RT-PCR analysis showed that *Zdhhc19* mRNA was only detectable in the testis. No signal was detected in the other eight tissues.

**Figure 2 ijms-22-08894-f002:**
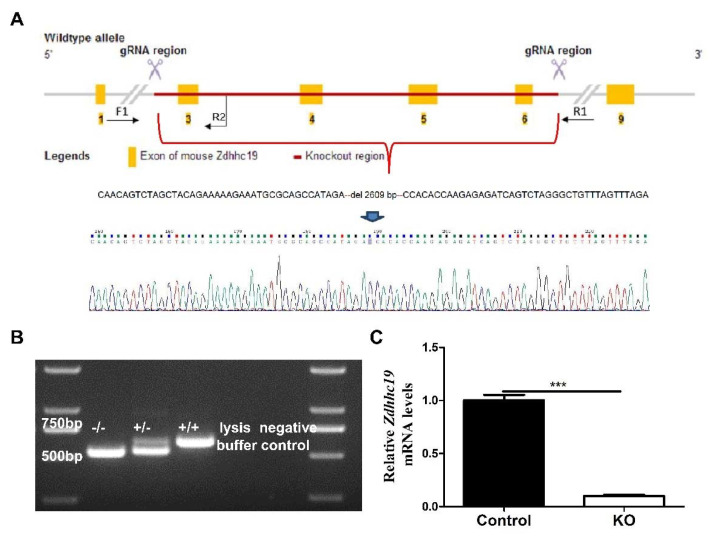
CRISPR/Cas9 strategy to generate *Zdhhc19* KO mice. (**A**) targeting scheme of *Zdhhc19* KO mice. Yellow boxes represent exons of the *Zdhhc19* gene on mouse chromosome 16, and black arrows (F1, F2 and R1) indicate the site of designed primers for genotyping. Red line indicates the target regions of sgRNAs. Exon 3–6 was deleted upon the injection of the mixtures of Cas9 and *Zdhhc19* sgRNA into zygotes. (**B**) PCR of wildtype (+/+), heterozygous (+/−), and homozygous (−/−). A 536bp band was amplified from the *Zdhhc19* KO allele using F1 and R1 primers, and a 617bp band was amplified from the wildtype allele using F1 and R2 primers. (**C**) *Zdhhc19* mRNA expression levels in control and *Zdhhc19* KO adult testes were examined by RT-qPCR. Data are presented as mean ± SEM. *** *p* < 0.001.

**Figure 3 ijms-22-08894-f003:**
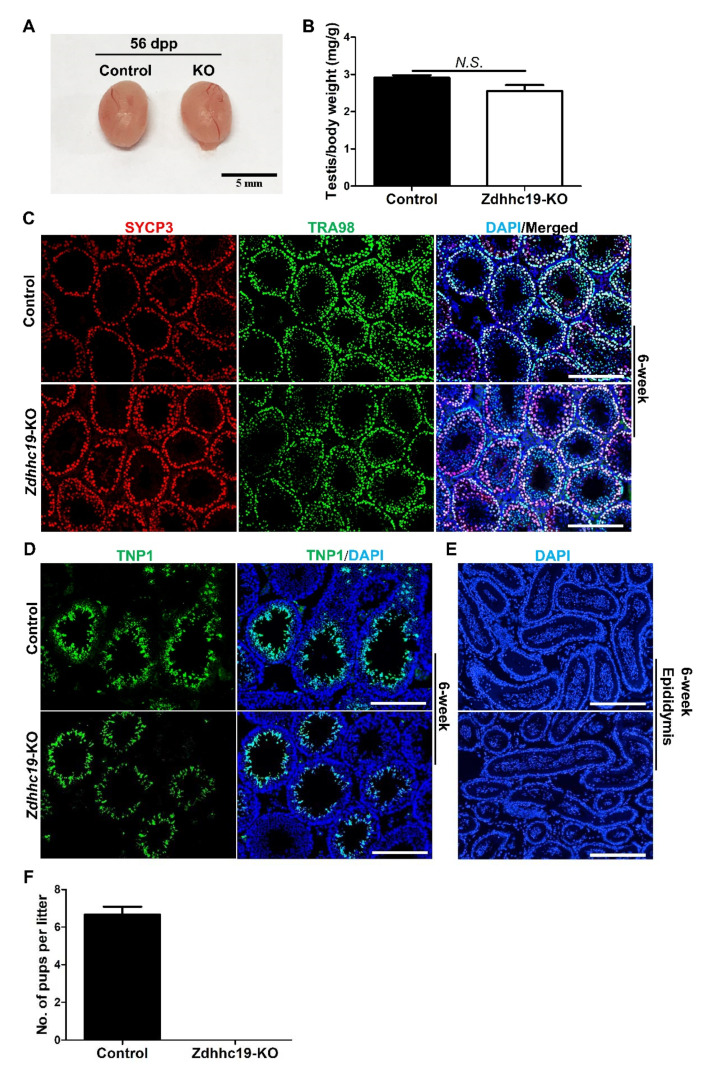
*Zdhhc19* deletion does not affect spermatogenesis, but leads to male infertility in mice. (**A**) Gross morphology of testes from control and KO mice at 56 dpp. (**B**) Averaged testis/body weight ratio was calculated from 4 control and 3 *Zdhhc19* KO littermates at 42 dpp. Data are presented as mean ± SEM, N.S.: no significance. (**C**) The development of pan-germ cells (by TRA98 staining) and spermatocytes (by SYCP3 staining) was examined by IHF on KO and control testis sections from 6-week mice. (**D**) The elongated spermatids were examined by IHF with a TNP1 antibody, counterstained with DAPI on KO and control testis sections from 6-week mice. (**E**) DAPI stained sections of epididymides from control and KO mice at 6-week. (**C**–**E**) Scale bars, 200 μm. (**F**) Average pups’ numbers per litter were calculated based on 6 litters from 3 control and 3 KO male mice, each of which was separately bred with 2 wildtype females for at least 2 months. Data are presented as mean ± SEM.

**Figure 4 ijms-22-08894-f004:**
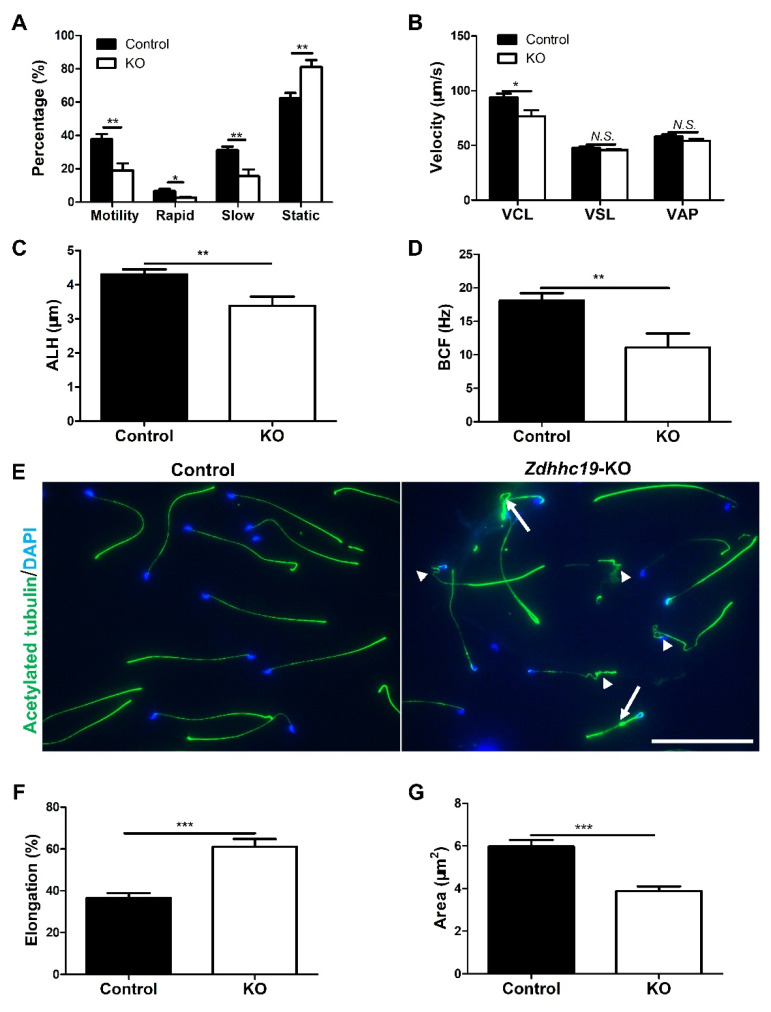
*Zdhhc19* deletion caused abnormal sperm motility and sperm flagella. (**A**) CASA analyses of sperm motility on *Zdhhc19* KO and control littermates. (**B**) Analyses of sperm velocity by CASA. VCL, curvilinear velocity; VSL, straight line velocity; VAP, average path velocity. (**C**) Analyses of sperm amplitude of lateral head displacement (ALH) by CASA. (**D**) Analyses of sperm beat/cross frequency (BCF) by CASA. (**E**) Immunofluorescence analyses of acetylated-TUBULIN on spermatozoa from adult KO and control littermates, counter-stained with DAPI. White arrows indicate sperm with bent tails, while arrowheads point to coiled sperm tails. Scale bars, 200 μm. (**F**,**G**) Analyses of sperm elongation (**F**) and sperm area (**G**) by CASA. (**A**–**D**,**F**,**G**) Data are represented as the mean ± SEM of three sperm samples per group. * *p* < 0.05; ** *p* < 0.01, *** *p* < 0.001; N.S.: no significance.

**Figure 5 ijms-22-08894-f005:**
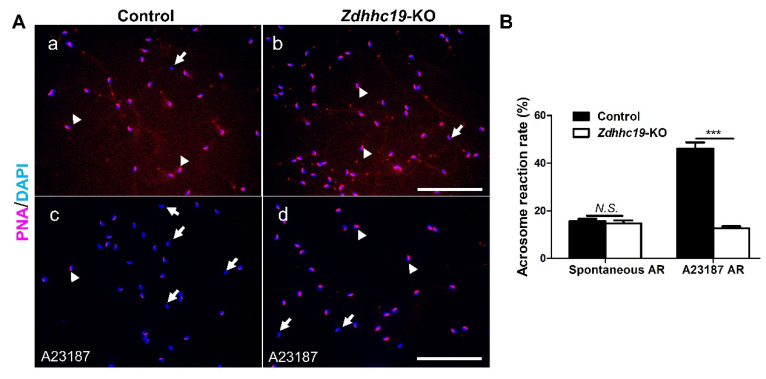
The induced acrosome reaction rate was significantly reduced upon *Zdhhc19* KO. (**A**) Sperm without (**a**,**b**) or with treatment of A23187 (**c**,**d**) were stained with Rhodamine-PNA (red) and counter-stained with DAPI (blue). The acrosome-reacted sperm had DAPI-labeled nuclei but lacked PNA staining (white arrows), whereas intact acrosomes were stained positive for both PNA and DAPI-labeled nuclei (white arrow heads). Scale bars, 200 μm. (**B**) The proportions of sperm that underwent spontaneous AR and induced AR were calculated from between *Zdhhc19* KO and control littermates. Data are represented as the mean ± SEM of three sperm samples per group. N.S.: no significance; *** *p* < 0.001.

**Figure 6 ijms-22-08894-f006:**
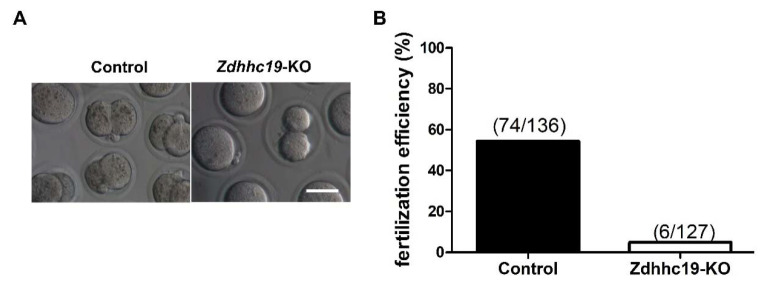
IVF analysis of *Zdhhc19* KO spermatozoa. (**A**) Cumulus-intact eggs were incubated in vitro with capacitated *Zdhhc19* KO or control sperm. Scale bar, 200 μm. (**B**) The percentage of 2-cell stage embryos upon IVF with control or KO sperm were calculated.

## Data Availability

The data presented in this study are available on request from the corresponding author.

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
