# Peer review of "ZDHHC19 Is Dispensable for Spermatogenesis, but Is Essential for Sperm Functions in Mice"

_ijms, 2021, doi:10.3390/ijms22168894_

Round 1

Reviewer 1 Report

This paper analyzes the function of ZDHHC19 in spermatogenesis and sperm function. By using ZDHHC19 KO mice, the authors show that while ZDHHC19 is not required for spermatogenesis, sperm function is abnormal after loss of this protein. Therefore, ZDHHC19 KO mice are infertile. The experiments are well done, and they thoroughly address the main questions. There are minor points that the authors need to address.

Minor points:

  • Figure 1. As qPCR data is depicted relative to a control level of expression, the authors need to explain which group in Fig. 1A and 1B are used as baseline control. The authors also need to mention in the figure legend the housekeeping gene (GAPDH).

  • Figure 2. The authors show that the knockout strategy works by using qPCR. Although not required, it would be informative to show loss of protein by Western blot if an antibody is available.

  • Some sentences need to be rewritten as they do not make sense; words are repeated back to back; adverbs are not used properly – lines 127-129, 196-197, 234-236, 286-290

  • The nomenclature of ZDHHC19 KO mice should be kept throughout the paper. Sometimes, the authors use the term ZDHHC19 mutants (lines 187, 210, 219)

  • The discussion section needs to be rewritten to exclude a rehash of the result section. Part of the 2nd paragraph and the entire 3rd paragraph are literally a second result section; note that there are zero references in paragraph 3 since this is not a discussion paragraph but a result paragraph.

Reviewer 2 Report

In this paper, Wang and colleagues analyzed the effects of Zdhhc19 gene deletion in mouse spermatogenesis. They found any morphological alteration in Zdhhc19 KO mice, as well as an intact spermatogenesis. However, many modifications in sperm function, including motility, acrosome reaction and fertilization ability, were recorded. So, the authors concluded that Zdhhc19 gene is essential for sperm physiology in mice.

The paper is interesting and well conducted, however, many revisions should be made prior its acceptance in IJMS:

- My main concern regards the discussion, since all the authors do is to recap the results, without giving any interpretation or proposing any function for Zdhhc19 gene in mouse spermatogenesis (the IHF for ZDHHC19 could help in this way);

- Also, the Material and Methods section lacks much information: the number of used animals, the sacrifice and collection of the tissues (in this regard, please in line 358, substitute killed with another word), the dilution of the used antibodies and of Rhodamine-PNA); moreover, in line 329, please specify the picture were processed for what;

-  In the Results, the authors should avoid “anticipation” of the discussion (as in lines 168-171 and 220-222); moreover, in Figure 3, a specific localization for TRA98 in the interstitial compartment is observable, the authors should justify this, finally, lines 196-198 should be rephrased (acrosome reaction is mentioned three times in three lines);

- To better explore the sperm abnormalities, to see whether head and tail defects are associated, a double staining with PNA and acetylated tubulin should be performed;

- In the Introduction, the authors should better clarify why decided to study just ZDHHC19 and not ZDHHC11.

Round 2

Reviewer 2 Report

The Authors addressed all the raised issues and the paper is worth of publication in IJMS in this form.